# Investigating latent mean differences in achievement emotions among Chinese secondary EFL learners: A gender and grade perspective

Kunbang Wang[1], Yajun Wu[2]*, Xia Kang[3,4]

1 School of Foreign Languages, Huazhong University of Science and Technology, Wuhan, China, 2 School of Humanities and Education, Foshan University, Foshan, China, 3 School of Mathematics and Information Science, Guangzhou University, Guangzhou, China, 4 Faculty of Education, the University of Hong Kong, Hong Kong, China

* wuyajun1225@163.com

**Data Availability Statement:** The data underlying the results presented in the study are available

## Abstract

The control-value theory (CVT) of achievement emotions posits that achievement emotions are significantly associated with the key indicators of academic outcomes, including academic motivation, engagement, and performance. Existing studies have tested the theoretical hypothesis of the CVT in a variety of cultures, disciplines, and samples. However, evidence is limited for whether there are gender and grade differences in achievement emotions, especially in the context of English as a Foreign Language (EFL). 1,460 Chinese secondary school students (male $N = 671$; female $N = 789$; seventh-graders $N = 731$; eighth-graders $N = 729$) took part in the study. Confirmatory factor analyses and multi-group analyses were conducted to explore the possible gender and grade differences in EFL-related achievement emotions. Results indicated that there are gender or grade differences in EFL-related enjoyment, anxiety, and boredom, while hope and pride did not. Both limitations and implications are discussed.

## Introduction

Achievement emotions refer to the emotions experienced in contexts of learning, classroom, and testing, which are closely related to achievement and achievement-related activities [1–3]. The significance of achievement emotions in subject education and students' well-being is increasingly valued by educators and researchers [4–6]. Students with higher positive achievement emotions (e.g., enjoyment, hope, and pride) are more likely to be motivated to engage in learning activities [3,7–9], adopt higher-level learning strategies [2,10], maintain superior self-regulation in learning [11], and obtain excellent academic achievement [12]. Besides, positive achievement emotions have also been shown to contribute to students' cognitive resources [13,14], life satisfaction, and subjective well-being [6,15]. In contrast, negative achievement emotions (e.g., boredom and anxiety) would create negative impacts on students' motivation, engagement, and academic achievement [16–18].

from Zenodo (https://doi.org/10.5281/zenodo.11095878).

**Funding:** Yajun Wu was awarded the fund by the 2023 Guangdong Philosophy and Social Science Foundation Special Projects (Grant number: GD23WZXY02-02).

**Competing interests:** The authors have declared that no competing interests exist.

Achievement emotions are woven into the multiple aspects of learning, however, their differential impact on different student groups should also be considered [19]. For instance, [20] documented that foreign language-related achievement emotions are dynamic and change as students age. Meanwhile, a meta-analysis by [21] focusing on gender differences in children's emotional expression found that girls showed more positive emotions than boys. The existing studies have confirmed gender and age differences in emotional experience and expression. However, the existing studies on achievement emotions from the perspective of gender or age comparison were mostly conducted in the Western context, and the limited research that took the domain specificity of achievement emotions into account has concentrated primarily on mathematics education [22,23], with very few studies being conducted in the EFL context [24]. To close this research gap, the present study measured and compared the latent mean differences in achievement emotions across gender and grade in a sample of 1,460 Chinese secondary EFL learners. The findings of this study contribute preliminary empirical evidence to educators in China and other countries eager to improve secondary EFL learners' academic and well-being outcomes through precise intervention in enhancing positive achievement emotions and ameliorating negative achievement emotions.

## Literature review

### Control-value theory of achievement emotions

According to the control-value theory of achievement emotions, students' subject-specific emotional experience is primarily determined by their cognitive control appraisal, value appraisal, and the interaction of control and value appraisals [3,12,25]. Further, achievement goals [26], teacher support [9], school psychological capital [27], and classroom social environment [28] were identified as the distal antecedents of achievement emotions for they were postulated to act indirectly on achievement emotions through control and value appraisals [3,13]. In addition, guided by the CVT [3], a number of studies have addressed the outcome variables of achievement emotions, such as engagement, learning strategies, academic achievement, and subjective well-being [2,6,7,12].

Drawing on the CVT, many empirical studies were conducted to verify the mediating role of achievement emotions (for a review, see [29]), however, related research was mostly carried out with a specific group of students (e.g., primary school students) as samples and ignored the minute differences within the sample (e.g., gender and grade differences). Specifically, although the whole sample might satisfy a particular theoretical or hypothetical model, attention should also be paid to the different types of samples for the nuances would provide policymakers and educators with more accurate information for implementing an intervention in achievement emotions.

### Gender differences in achievement emotions

There are gender differences in emotional understanding [30], that is, the emotional footprint experienced by boys and girls might be different as they could be different in perceived control and value appraisals [3]. For example, [23] compared mathematics-related academic emotions across genders and found that girls experienced less enjoyment and pride and more anxiety, hopelessness and shame than boys in learning mathematics. A meta-analysis by [21] concentrating on gender differences in children's emotion expression found that girls were higher than boys in positive emotions and internalizing emotions such as anxiety, sadness, and sympathy, but lower than boys in external emotions such as anger. In the EFL context, gender differences in enjoyment and anxiety were also found [31,32]. Specifically, girls reported higher levels of enjoyment and anxiety than boys.

Studies have documented gender differences in emotions, however, there are at least three deficiencies in existing studies. First, measurement invariance was rarely tested, which might result in biased estimates [33]. Second, *t*-test and ANOVA were the main methods adopted to make comparisons across groups, which could not disattenuate from possible measurement error [34]. Third, EFL researchers mainly addressed enjoyment and anxiety [20,32,35], which neglected the rich diversity of achievement emotions [2,13]. The present study was designed to make up for the research deficiencies mentioned above.

## Grade differences in achievement emotions

Grade differences in achievement emotions assume that individuals' academic emotions would develop dynamically with grades. For example, [36] explored the relationships between achievement emotions and grade levels and found that the representation of achievement emotions is a function of grade level. Similarly, [37] documented that students' positive achievement emotions (i.e., enjoyment and pride) would decline while their negative achievement emotions (i.e., anxiety, boredom, and hopelessness) would increase after transitioning from elementary school to secondary school. In addition, [38] also found that the academic emotions of students with learning difficulties were more likely to deteriorate into negative ones at the turn between primary school and lower secondary school.

The limited cross-grade comparison of achievement emotions manifested that students' achievement emotions vary across different school levels. However, the existing studies have some deficiencies that need to be fulfilled. First, the existing studies were majorly conducted in Western contexts, with Eastern participants being neglected. Second, achievement emotions associated with literacy and mathematics were studied, violating the domain-specificity of achievement emotions [22,39]. Third, traditional methods of ANOVAs were performed when comparing the achievement emotions across school grade levels, which could not avoid possible measurement errors [40]. Given these limitations, this study focused on the achievement emotions of secondary EFL learners in China and investigated the possible latent mean differences across grades seven and eight.

## The present study

As proposed, theoretically and empirically, achievement emotions are correlated with a broad set of indicators of academic and well-being outcomes. However, few studies have tested the possible latent mean differences of achievement emotions across gender and grade level, especially the achievement emotions in EFL setting. Accordingly, this study aimed to address the following four research questions:

1. What is the factor structure of the Achievement Emotion Questionnaire (AEQ) in a Chinese secondary EFL context?

2. Does the measurement invariance of EFL-related achievement emotions (i.e., enjoyment, pride, hope, anxiety, and boredom) across gender and grade level established?

3. Are there significant differences in the latent means of EFL-related achievement emotions across gender? If so, which discrete achievement emotions show significant difference?

4. Are there latent mean differences across grade level in EFL-related achievement emotions? If so, which discrete achievement emotions are significantly different between grades seven and eight?

## Methods

### Participants and procedure

**Ethics statements.** The present study was approved by the Human Research Ethics Committee of the University of Hong Kong (Reference No.: EA 2003020). Additionally, prior to the questionnaire survey, we collected written informed consent forms signed by the participants and verbal informed consent from the participants' parents or legal guardians. Only data obtained with informed consent forms were collected and analyzed.

**Participants.** Stratified random sampling method was employed to select three secondary schools and their 1,460 students. These three secondary schools are from Yunnan province, China. Participants were in Grades 7 (the first year of secondary schooling) and 8 (the second year of secondary schooling), with a mean age of 13.46 years (*SD* = .74). Specifically, the sample consisted of 731 seventh graders and 729 eighth graders (female n = 789, male n = 671). Both the participants and their English teachers provided the written informed consent before conducting the questionnaire survey. Besides, verbal informed consent was also obtained from participants' parents or legal guardians.

**Procedure.** First, the researchers contacted the school leaders and EFL teachers and enlisted their cooperation in taking part in the questionnaire survey. Second, the entire questionnaire survey lasted about two months, starting on April 20, 2021 and ending on June 25, 2021. Third, with the help of EFL teachers, the questionnaire survey was administered during class hours in learning English. The purpose of the questionnaire survey, as well as the matters needing attention in answering the questionnaire, were explained to participants by the EFL teachers. Meanwhile, participants were given sufficient freedom to stop or withdraw the questionnaire at any time without causing any adverse effects. In total, the questionnaire took 10–15 minutes. Fourth, only the completed surveys with written and verbal informed consent were included in the analysis.

### Measure

**EFL-related achievement emotions questionnaire.** Participants' achievement emotions that experienced in learning English were measured by the questionnaire adapted from [41]. Achievement emotions are domain-specific [22], whereas [41] questionnaire is domain-general and does not focus on a specific subject. Therefore, all items of the original AEQ were adapted to be relevant to the EFL context. First, in terms of situational context, achievement emotions would vary in the academic settings of attending class, doing homework, and taking tests and exams [41]. This study focused on five classroom-related achievement emotions: enjoyment, pride, hope, anxiety, and boredom for they were identified as the most commonly experienced emotions in achievement settings [17,42,43]. Specifically, enjoyment, hope, pride, and boredom were measured with four items each, while anxiety was measured by three items. In total, we drew nineteen items from the original AEQ and the sample items are: "I enjoy the challenge of learning English" (enjoyment); "I feel confident when studying English" (hope); "I think I can be proud of my accomplishments at studying English" (pride); "I get tense and nervous while studying English" (anxiety); "While studying English I seem to drift off because it's so boring"(boredom).

Fig 1 presents the correlated five-factor structure of the EFL-related AEQ. Each item of the EFL-related AEQ was rated on a five-point Likert scale (1 = strongly disagree to 5 = strongly agree). Higher scores indicate stronger intensity of achievement emotions. Item factor loadings ranged from .40 to .87 and were significant at $p < .001$. The Cronbach's alphas of the five subscales of EFL-related AEQ ranged from .62 to .88 (see Table 1), indicating that the reliability

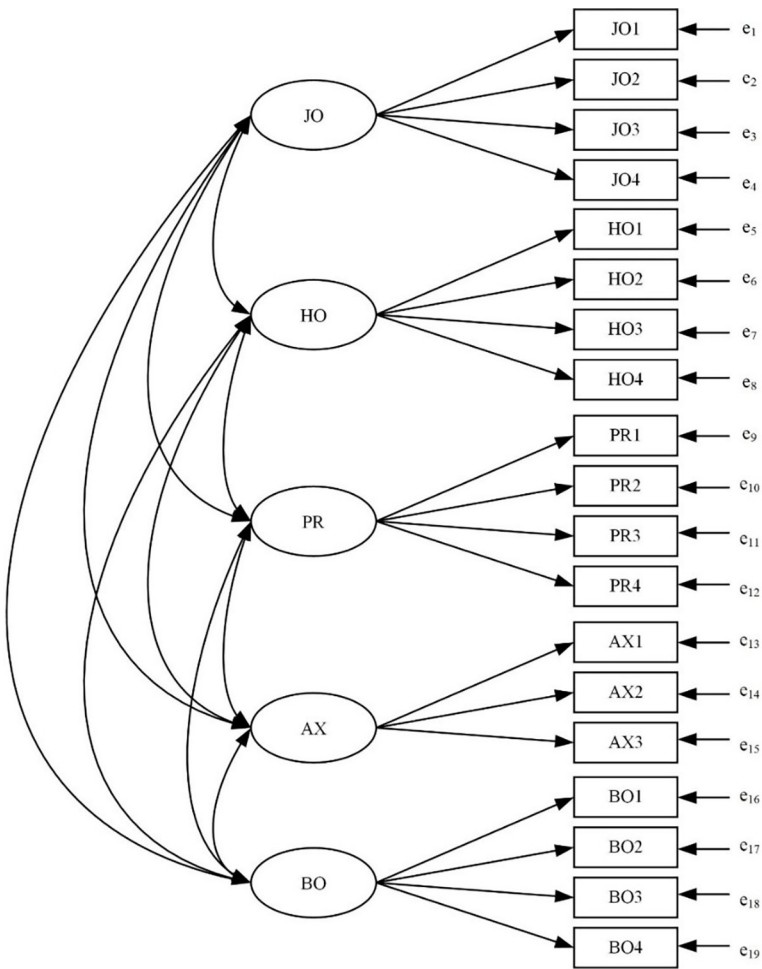

**Fig 1. Confirmatory factor model.** *Note.* JO = Enjoyment; HO = Hope; PR = Pride; AX = Anxiety; BO = Boredom.

of these subscales was acceptable or good. Besides, the fit of the baseline model depicted in Fig 1 for boys, girls, seventh-grade, and eighth-grade was good (see Table 2). Moreover, participants' demographic information including age, gender, and grade was also collected.

## Data analysis

Data were analyzed in three stages using *Mplus* 8.3 [44]. First, descriptive statistics was manifested and internal consistencies of each subscale of EFL-related AEQ were verified. In addition, the internal consistencies of each subscale were also separately assessed by gender and grade level. Second, confirmatory factor analyses (CFA) were conducted to substantiate the validity of each construct. Third, multi-group CFA was performed to investigate the measurement invariance and the possible latent mean differences of the EFL-related AEQ across gender and grade level.

## Results

### Preliminary analysis

The item quality of the 19-item EFL-related AEQ was firstly analyzed and evaluated by measuring the skewness and kurtosis values of each item. The values of skewness ranged from -.36

**Table 1. Factor loadings in the CFA for latent variables in all samples.**

| Items | Factor loadings | | | |
|---|---|---|---|---|
| | Gender | | Grade | |
| | boys | girls | seventh | eighth |
| **Enjoyment (JO)** | | | | |
| JO1 | .73 | .73 | .77 | .70 |
| JO2 | .82 | .82 | .83 | .80 |
| JO3 | .76 | .75 | .77 | .76 |
| JO4 | .83 | .84 | .82 | .85 |
| **Hope (HO)** | | | | |
| HO1 | .76 | .75 | .76 | .75 |
| HO2 | .79 | .77 | .74 | .83 |
| HO3 | .77 | .73 | .77 | .73 |
| HO4 | .82 | .82 | .82 | .82 |
| **Pride (PR)** | | | | |
| PR1 | .79 | .75 | .72 | .80 |
| PR2 | .78 | .77 | .74 | .80 |
| PR3 | .82 | .75 | .79 | .79 |
| PR4 | .74 | .79 | .72 | .80 |
| **Anxiety (AX)** | | | | |
| AX1 | .55 | .54 | .55 | .53 |
| AX2 | .80 | .87 | .86 | .84 |
| AX3 | .45 | .40 | .40 | .41 |
| **Boredom (BO)** | | | | |
| BO1 | .69 | .80 | .75 | .73 |
| BO2 | .77 | .80 | .76 | .81 |
| BO3 | .70 | .68 | .71 | .67 |
| BO4 | .67 | .78 | .71 | .74 |

*Note*. Factor loadings are standardized coefficients. All factor loadings are statistically significant at $p < .001$.

to .80 and the values of kurtosis ranged from -.81 to .24, which indicated that the distributional properties of all studied items were normal for both skewness and kurtosis ranged from -2 to 2 [45]. The normal distribution of data justified the use of maximum likelihood (ML) estimation in assessing the model parameters and fit indices. The means ranged from 2.08 to 3.50 and the standard deviations ranged from .92 to 1.19, indicating a fairly narrow spread of scores around their mean.

The coefficient alpha reliabilites for the EFL-related AEQ as well as the five subscales were calculated across subsamples (e.g., male and female subscamples). The coefficients of the

**Table 2. Fit indices for CFA of EFL-related AEQ across gender and grade.**

| | $\chi^2$ | df | $\chi^2/df$ | CFI | TLI | RMSEA (90% C.I.) | SRMR |
|---|---|---|---|---|---|---|---|
| **Gender** | | | | | | | |
| Boys | 426.408 | 142 | 3.002 | 0.958 | 0.949 | .055(.049,.061) | 0.038 |
| Girls | 643.645 | 142 | 4.533 | 0.939 | 0.926 | .067(.062,.072) | 0.048 |
| **Grade** | | | | | | | |
| Seventh grade | 622.532 | 142 | 4.384 | 0.936 | 0.922 | .068(.063,.074) | 0.044 |
| Eighth grade | 501.259 | 142 | 3.53 | 0.952 | 0.942 | .059(.053,.065) | 0.043 |

**Table 3. Manifest intercorrelations and internal consistency of the EFL-related AEQ across gender and grade groups.**

| | | | | | | | Cronbach's alpha | |
|---|---|---|---|---|---|---|---|---|
| | JO | HO | PR | AX | BO | Total | Boys | Girls |
| **Gender** | | | | | | | | |
| JO | - | .83 | .73 | -.19 | -.58 | .87 | .86 | .87 |
| HO | .79 | - | .73 | -.19 | -.52 | .86 | .87 | .85 |
| PR | .59 | .71 | - | -.19 | -.38 | .86 | .86 | .85 |
| AX | -.32 | -.35 | -.32 | - | .43 | .62 | .64 | .65 |
| BO | -.64 | -.55 | -.36 | .45 | - | .83 | .80 | .85 |
| | | | | | | | Cronbach's alpha | |
| **Grade** | | | | | | | Seventh | Eighth |
| JO | - | .81 | .65 | -.31 | -.66 | | .88 | .86 |
| HO | .81 | - | .72 | -.30 | -.59 | | .85 | .86 |
| PR | .65 | .72 | - | -.27 | -.41 | | .83 | .87 |
| AX | -.31 | -.30 | -.27 | - | .50 | | .65 | .63 |
| BO | -.66 | -.59 | -.41 | .50 | - | | .82 | .83 |

*Note*. The correlations above the diagonal are for the boy sample and seventh-grade sample, respectively, and the correlations below the diagonal are for the girl sample and eighth-grade sample. All correlations are significant at $p < .01$.

subscales of enjoyment, hope, pride, and boredom ranged from .80 to .87 for the boy group, ranged from .83 to .87 for the girl group, ranged from .82 to .88 for seventh graders, and ranged from .83 to .86 for eighth graders, indicating that these four subscales possessed good reliability in the above four subsamples. Also, the Cronbach's alphas of anxiety subscale were higher than .62, which mean that anxiety subscale had moderate reliability in the four mentioned subsamples [46] (see Table 3).

Further, the intercorrelations for the manifest variables of enjoyment, hope, pride, anxiety, and boredom were also presented in Table 3. Enjoyment, hope, and pride are the most commonly experienced positive achievement emotions [3,41,43] and they were positively intercorrelated in all subsamples. Anxiety and boredom are the two prominent negative achievement emotions in EFL settings [47,48] and they were also positively intercorrelated with each other. However, the correlations between the positive and negative achievement emotions were moderately negative (see Table 3 for more details).

## Confirmatory factor analysis

Several goodness-of-fit indices were used to assess the fit of the CFA model: chi-square ($\chi^2$), Tucker-Lewis index (TLI), comparative fit index (CFI), root mean square error of approximation (RMSEA), and standardized root mean square residual (SRMR) [49]. Given that $\chi^2$ statistics is sensitive to sample size, the ratio of $\chi^2$ to its degree-of-freedom ($\chi^2/df$) was also evaluated. [50] documented that the $\chi^2/df$ value less than 5 indicated that the fit between the hypothetical model and sample data was acceptable. Traditional cutoff criteria were taken as evidence of excellent and adequate fit: (a) CFI$\geq$ .95 and $\geq$.90, respectively (b) TLI $\geq$.95 and $\geq$.90 respectively; (c) RMSEA $\leq$.06 and $\leq$.08; and (d) SRMR $\leq$.08 and $\leq$.10 [51,52].

## Measurement invariance testing

Multi-group CFA model was run to examine the measurement invariance of EFL-related AEQ across gender and grade groups. In establishing measurement invariance, researchers used a series of increasingly restrictive steps, that is, configural, metric and scalar invariance was

**Table 4. Fit indices for measurement invariance across gender and grade groups.**

| Model | $\chi^2$ | df | $\chi^2/df$ | CFI | ΔCFI | TLI | RMSEA (90%C.I.) | SRMR |
|---|---|---|---|---|---|---|---|---|
| M1[a]: Configural invariance | 1070.053 | 284 | 3.768 | .947 | - | .937 | .062(.058, .066) | .044 |
| M2[a]: Metric invariance | 1097.775 | 298 | 3.684 | .946 | .001 | .938 | .061(.057, .065) | .046 |
| M3[a]: Scalar invariance | 1219.846 | 317 | 3.848 | .939 | .007 | .935 | .063(.059, .066) | .053 |
| M4[b]: Configural invariance | 1123.790 | 284 | 3.957 | .944 | - | .932 | .064(.060, .068) | .043 |
| M5[b]: Metric invariance | 1149.543 | 298 | 3.858 | .943 | .001 | .935 | .063(.059, .067) | .047 |
| M6[b]: Scalar invariance | 1193.012 | 317 | 3.763 | .941 | .002 | .937 | .062(.058, .065) | .050 |

*Note.*

[a] Fit indices for measurement invariance tests of the model across genders.

[b] Fit indices for measurement invariance tests of the model across grade level.

examined in sequence. The aim of this was to compare the mean differences of five discrete foreign language learning-related achievement emotions across groups, therefore, scalar invariance is required [53]. Measurement invariance was established if (1) the overall model fit is acceptable [54] and (2) changes in the comparative fit index (ΔCFI) between two nested models ≤ 0.01 [55].

Given that configural invariance is commonly viewed as the baseline model and is the necessary condition for testing invariance of measurement parameters, thus, the configural invariance of the EFL-related AEQ was firstly assessed across age and grade groups. As shown in Table 4, measurement invariance of EFL-related AEQ was established. Specifically, in M1[a] and M1[b], CFI ≥.90, TLI ≥.90, RMSEA ≤.08, and SRMR ≤.08, indicating that the overall model fit is adequate, thus, the configural invariance across gender and grade groups was supported. The establishment of configural invariance means that the pattern of fixed and non-fixed parameters hold up similarly for all groups examined. The overall factor structure of EFL-related AEQ fits well for all gender and grade groups, which lays foundation for assessing metric invariance [49].

Next, metric invariance which aims to assess whether the factor loadings are equivalent across the groups was examined. With this step, all factor loadings were constrained to be equal across all the groups examined. ΔCFI in M2[a] and M5[b] takes a value of .001, which is lower than the cutoff criteria of .01. Besides, based on the cutoff criteria above, the model fits were adequate in M2[a] and M5[b], indicating that metric invariance was established across gender and grade groups. As such, scalar invariance across the groups would be examined. Lastly, factor loadings and intercepts of all items were constrained to be equal across groups (M3[a] and M6[b]). As shown in M3[a], the model fit was acceptable and ΔCFI = .007 (meeting the cutoff criteria of less than .01), showing that scalar invariance exists across gender groups. Meanwhile, M6[b] demonstrates that the model fit was acceptable and ΔCFI satisfied the cutoff criteria, suggesting that scalar invariance also exists grade groups. In sum, the results in Table 4 indicated that measurement invariance exist across gender and grade groups.

## Structured latent mean differences across gender and grade

Based on the establishment of measurement invariance across both gender and grade levels, latent mean differences across these groups could be compared because measurement invariance ensures that cross-groups comparisons are meaningful and valid [49]. Compared with the traditional methods of *t*-test or ANOVA, the advantage of latent mean differences are that they could provide error-free measures of the latent variables by accounting for the random error of measurement for the observed variables associated with each latent variable [56]. In

**Table 5. Structural latent mean differences across gender and grade on EFL-related AEQ.**

|  | Constructs | | | | |
|---|---|---|---|---|---|
|  | JO | HO | PR | AX | BO |
| **Boys (reference)** | .000 | .000 | .000 | .000 | .000 |
| Girls |  |  |  |  |  |
| Mean estimate | .190 | .066 | -.015 | -.054 | -.271 |
| Standard error | .041 | .042 | .047 | .035 | .044 |
| Test statistic | 4.578 | 1.574 | -.315 | -1.554 | -6.123 |
| *p*-value | **.000** | .115 | .753 | .120 | **.000** |
| **Seventh grade (reference)** | .000 | .000 | .000 | .000 | .000 |
| Eighth grade |  |  |  |  |  |
| Mean estimate | -.054 | -.052 | -.061 | .068 | .203 |
| Standard error | .041 | .041 | .047 | .035 | .043 |
| Test statistic | -1.314 | -1.266 | -1.313 | 1.955 | 4.683 |
| *p*-value | .189 | .205 | .189 | **.051** | **.000** |

the comparisons of latent mean differences across gender groups, boys were treated as the reference group. While in the comparisons of latent mean differences across grade levels, seventh graders served as the reference group. The latent mean of these two reference groups was constrained to zero. The other two groups (i.e., girls and eighth graders) were regarded as comparison groups, with their latent mean being freely estimated.

The structural latent mean differences across gender and grade level were presented in Table 5. First, the fit statistics for the latent mean structures across gender groups were good: $\chi^2$ (312) = 1155.201, $\chi^2/df$ = 3.703, CFI = .943, TLI = .938, RESEA = .061 90% C. I. (.057, .065), SRMR = .047. Results showed that the mean of EFL-related enjoyment for girls was significantly higher than that for boys ($p < .001$) and the mean of EFL-related boredom was significantly lower than that for boys ($p < .001$). In addition, there were no structured means differences between boys and girls in the achievement emotions of hope, pride, and anxiety. Second, the fit statistics for the latent mean structures across grade level were also good: $\chi^2$ (312) = 1165.465, $\chi^2/df$ = 3.735, CFI = .943, TLI = .937, RESEA = .061 90% C.I. [.058,.065], SRMR = .047. Eighth graders had higher mean scores than seventh graders on boredom ($p < .001$) and anxiety ($p < .05$). However, no significant differences were obtained between the seventh and eighth graders on enjoyment, hope, and pride.

## Discussion

Given the importance of achievement emotions to academic and well-being outcomes such as academic engagement, learning motivation, learning satisfaction and academic performance [12,15,17,57], an increasing number of studies have begun to explore the profile of students' achievement emotions. However, cross-group comparative studies on achievement emotions are relatively insufficient. The few studies were confined to the mathematics domain [23], and *t*-test or ANOVA methods were chiefly used, which could not avoid the possible measurement error [34]. To fill these gaps, the present study tested the factor structure and measurement invariance of EFL-related AEQ across gender and grade groups in a sample of 1,460 Chinese secondary school students. Subsequently, the mean scores of achievement emotions across gender and grade levels were separately examined. As discussed hereafter, there are similarities and differences between the findings of the present study and those of previous studies, and they may also provide practical guidance for emotionally beneficial education.

## Factor structure of the EFL-related AEQ in a sample of Chinese secondary school students

This study found that the 19-item EFL-related AEQ with good psychometric properties measures secondary school students' enjoyment, hope, pride, anxiety, and boredom in the Chinese EFL classroom context. This finding is consistent with the results of past research examining the validity of AEQ in the EFL context [12,58,59]. This finding contributes to the literature in two ways. First, this study took Chinese secondary EFL learners as participants, which would further expand the application scope of the AEQ. Second, the EFL-related AEQ in the present study measures more achievement emotions, beyond enjoyment and anxiety in one single investigation [17,20,60]. There are as many as nine achievement emotions commonly experienced by students in an academic context [1,3], however, existing studies in the field of EFL education are mostly limited to examining a single emotion [8,61] or two or three emotions [17,47,62]. The EFL-related AEQ in the present study includes five discrete achievement emotions, which provide a reliable and valid instrument for measuring a broader range of EFL-related achievement emotions in a classroom situation.

## Measurement invariance of the EFL-related AEQ across gender and grade

This study also found that the EFL-related AEQ was strongly invariant across gender and grade, indicating that this questionnaire could be used for comparison of achievement emotions across gender and grade. This finding is consistent with previous studies on the measurement invariance of the AEQ within different subject areas [58,63,64]. However, none of the previous studies explored the factorial invariance of AEQ across gender and grade in the Chinese EFL classroom context. This study contributes to filling this research gap in the literature by examining the configural, metric, and scalar invariance of EFL-related AEQ across gender and grade in a sample of Chinese secondary EFL learners [53], which lays the groundwork for cross-group comparisons of achievement emotions.

## Latent mean differences across gender in EFL-related achievement emotions

It was found that girls experienced higher levels of enjoyment and lower levels of boredom than boys, while there were no significant differences in hope, pride, and anxiety. This finding was partially consistent with the existing studies on gender differences in AEQ [31,65,66]. Precisely, the similarities and differences between this finding and those of existing studies are reflected in the following aspects. First, gender differences in achievement emotions were also confirmed in the Chinese EFL context, implying that teachers need to take gender differences into account when boosting the facilitative effects of EFL learners' positive achievement emotions or alleviating the debilitative effects of negative achievement emotions. Second, compared with boy students, girl students had significantly higher mean enjoyment scores and lower mean boredom scores in the EFL classroom settings, which is similar to findings in [65]. The possible reason is that EFL learning is one feminine domain, girl students would be more desirable to devote themselves to EFL learning [67]. Third, no significant mean differences in EFL-related pride, hope, and anxiety were found between boys and girls, which differs from the findings of [31]. [31] found that female students experienced more pride and anxiety than male students in the foreign-language classroom context. Given that participants' achievement emotions are moderated by age, culture, and socioeconomic status (SES) [3,68,69], the differences in students' achievement emotions might be caused by differences in age, cultural backgrounds, and SES.

### Latent mean differences across grade in EFL-related achievement emotions

In EFL learning, eighth graders experienced more anxiety and boredom than seventh graders. However, there were no significant differences in positive achievement emotions (i.e., enjoyment, pride, and hope) across grade levels. These findings are consistent with previous studies [36,37,70]. For example, Vierhaus et al. (2016) reported that students' enjoyment decreased, whereas boredom increased from Grade five to Grade seven. Several possible reasons for grade differences in achievement emotions related to EFL learning exist. First, the decline in control and value appraisals is the possible reason for the increase in negative achievement emotions. According to the CVT, control and value appraisals are the two proximal antecedents of achievement emotions [3,12]. Furthermore, [71] documented that students' English expectancies and values declined, whereas the difficulty of English courses increased continuously from Grade seven to Grade eleven. Therefore, it could be inferred that students' negative achievement emotions would be increased while the positive achievement emotions hardly arise. Second, the "put exams first" mentality dominates both EFL learners and teachers in the Chinese context [72], implicating that students' anxiety levels would increase throughout secondary education as the senior high school entrance examination draws nearer. Meanwhile, the task difficulty of English learning in Grade eight is more difficult than in Grade seven [71], which would lead to an increase in boredom for task difficulty is positively correlated with boredom [73]. Lastly, in the Chinese cultural context, suppression of positive emotions, including enjoyment, hope, and pride, was highly valued for the purpose of maintaining interpersonal harmony [74]. Grade differences in EFL-related achievement emotions found in the present study suggested that educators intervene early to block the maladaptive development of students' achievement emotions and take steps to boost their positive achievement emotions.

## Limitation and suggestions for future research

The psychometric properties, measurement invariance, and latent mean differences of the EFL-related AEQ were explored in a sample of 1,460 Chinese secondary EFL learners. In addition to good psychometric properties and measurement invariance of EFL-related AEQ across gender and grade level, latent mean differences across gender and grade level were also obtained. However, three limitations need to be addressed. First, the EFL-related AEQ used in the present study included five but not all types of discrete achievement emotions (i.e., shame, hopelessness, anger, and relief) [3,41]. Future studies are recommended to consider all discrete achievement emotions to draw a comprehensive picture of the profile of students' achievement emotions in the EFL learning context. Second, several reasons that might account for gender or grade differences in achievement emotions were discussed, however, relevant empirical studies have not been carried out. Future research could further explore the reasons for the gender and grade differences in achievement emotions related to EFL learning through empirical research design.

The third limitation is that participants in this study involved only seventh- and eighth-grade secondary school students and thus the findings would be of limited value if they were outside this population. Therefore, by including all the three grades of secondary school (Grade seven, eight, and nine) in future research a more comprehensive understanding of the psychometric properties, measurement invariance, and gender and grade differences in EFL-related AEQ in a sample of secondary school students could be obtained.

## Educational implications

EFL educators would be advised that there were significant latent mean differences in students' achievement emotions across gender and grade level. Specifically, girls experienced more

enjoyment and less boredom in EFL learning than boys. And the levels of negative emotions, including anxiety and boredom, rose from Grade seven to Grade eight. Hence EFL educators who aim to develop a program to improve students' positive achievement emotions (e.g., enjoyment, hope and pride) and reduce their negative achievement emotions (e.g., anxiety and boredom) are suggested to be sensitive to gender and grade differences. Given that boys' achievement emotions in EFL classes were more negative than those of the girls, educators are advised to take measures, such as offering topics that boys are interested in and familiar with [75], to enhance boys' enjoyment and reduce their boredom in the EFL classroom setting. Also, negative achievement emotions related to EFL learning increased significantly from seventh to eighth grades, suggesting that EFL educators need to relieve students of boredom and anxiety. Balancing the difficulty levels of learning content with students' learning competence, constructing positive relationships with students, and making skillful use of humor [76,77] are the possible measures that teachers can take to alleviate anxiety and boredom among students.

## Conclusion

Achievement emotions have become one critical cornerstone and greatly influence the key indicators of academic outcomes in the EFL setting. The present study found that there were gender differences in enjoyment and boredom and grade differences in boredom and anxiety of Chinese secondary EFL learners. Besides, this study tested the psychometric properties and the measurement invariance of the EFL-related AEQ, supporting the validity and utility of the EFL-related AEQ in cross-group comparison. Any effort to create an emotionally supportive environment requires taking profiles of students' EFL-related achievement emotions into account to boost the beneficial effects of positive achievement emotions and diminish the detrimental effects of negative achievement emotions for teaching and learning EFL successfully.

## Supporting information

**S1 File.**
(SAV)

**S1 Raw data.**
(SAV)

## Acknowledgments

We appreciate the valuable comments and suggestions from the editor and reviewers. Additionally, we extend our gratitude to the school principal, English teachers, and EFL learners for their support and participation in the survey.

## Author Contributions

**Conceptualization:** Kunbang Wang, Yajun Wu, Xia Kang.

**Formal analysis:** Xia Kang.

**Funding acquisition:** Yajun Wu.

**Investigation:** Kunbang Wang, Xia Kang.

**Methodology:** Kunbang Wang, Yajun Wu.

**Project administration:** Yajun Wu.

**Validation:** Xia Kang.

**Visualization:** Xia Kang.

**Writing – original draft:** Kunbang Wang.

**Writing – review & editing:** Yajun Wu, Xia Kang.

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
