## [Decision Letter · Decision Letter 0]

10 Apr 2024

PONE-D-23-28676Investigating Latent Mean Differences in Achievement Emotions among Chinese Secondary EFL Learners: A Gender and Grade PerspectivePLOS ONE

Dear Dr. Wu,

Thank you for submitting your manuscript to PLOS ONE. After careful consideration, we feel that it has merit but does not fully meet PLOS ONE’s publication criteria as it currently stands. Therefore, we invite you to submit a revised version of the manuscript that addresses the points raised during the review process.

We look forward to receiving your revised manuscript.

Kind regards,

Alexander Rabadi

Academic Editor

PLOS ONE

Journal Requirements:

"Yajun Wu was awarded the fund by the 2023 Guangdong Philosophy and Social Science Foundation Special Projects (Grant number: GD23WZXY02-02)."               

Additional Editor Comments:

Thank you for submitting to PLOS ONE journal. Upon reviewing the manuscript and inspecting the methodology and prior literature review, the decision to proceed with the manuscript was made.

---

## [Author Response · Author response to Decision Letter 0]

1 May 2024

Dear Editor and Reviewers,

Thank you for your encouragement and giving us the opportunity to continue revising. We have re-read the submission format requirements and made modifications based on your editing suggestions. We received 5 editing suggestions in your previous email, and we have made the necessary changes and responded to them. We look forward to responding and revising seriously and responsibly. However, if there is any misunderstanding due to communication, please let us know directly. We will be grateful for your feedback and make the necessary improvements to enhance the quality of the manuscript.

1. Please ensure that your manuscript meets PLOS ONE's style requirements, including those for file naming. The PLOS ONE style templates can be found at https://journals.plos.org/plosone/s/file?id=wjVg/PLOSOne_formatting_sample_main_body.pdf andhttps://journals.plos.org/plosone/s/file?id=ba62/PLOSOne_formatting_sample_title_authors_affiliations.pdf

Responses: Thank you for your valuable comments and suggestions. We re-read the formatting requirements outlined by PLOS One and made appropriate adjustments to our manuscript to align with those standards. Specifically, Further modifications have been made to the format of the title page, page numbers, line spacing, table format, ensuring figures do not appear in the main text, and the placement of supporting information.

Responses: We fully endorse the concept of sharing data to advance scientific research. In addition to uploading supplementary files to the submission system, following your suggestion, we have also deposited the raw data in the Zenodo repository. You can access the data at https://doi.org/10.5281/zenodo.11095878. Through Zenodo, we make our data available to the public without any restrictions.

3. Thank you for stating the following financial disclosure: "Yajun Wu was awarded the fund by the 2023 Guangdong Philosophy and Social Science Foundation Special Projects (Grant number: GD23WZXY02-02)." Please state what role the funders took in the study. If the funders had no role, please state: "The funders had no role in study design, data collection and analysis, decision to publish, or preparation of the manuscript. If this statement is not correct you must amend it as needed. Please include this amended Role of Funder statement in your cover letter; we will change the online submission form on your behalf.

Responses: The corresponding author, Yajun Wu, received financial support for this study. On the one hand, the funding was provided by an organization rather than an individual, primarily for expenses such as printing paper questionnaires and covering travel costs for researchers. The funding organization did not participate in the study design, data collection and analysis, decision to publish, or manuscript preparation. On the other hand, the funding agency requires the disclosure of this funding information. All aspects of the paper’s design and writing are disclosed in the section of “Author contributions”.

Responses: Thank you for your valuable suggestions. Both ethics approval and written consent were obtained for the present study. According to your suggestions, the ethics statement was included in the section of “Methods”.

Responses: Considering your suggestion and the benefits of data sharing for academia and readers, I and all the co-authors of this study decided to share all the data used in this study through the Zenodo platform (https://doi.org/10.5281/zenodo.11095878). We will also adhere to data sharing in the future to better serve readers and academia. Thank you for your valuable comments and suggestions.

---

## [Editor Report · Decision Letter 1]

6 May 2024

Investigating Latent Mean Differences in Achievement Emotions among Chinese Secondary EFL Learners: A Gender and Grade Perspective

PONE-D-23-28676R1

Dear Dr. Wu,

We’re pleased to inform you that your manuscript has been judged scientifically suitable for publication and will be formally accepted for publication once it meets all outstanding technical requirements.

Kind regards,

Alexander Rabadi

Academic Editor

PLOS ONE
---

## [Editor Report · Acceptance letter]

21 May 2024

PONE-D-23-28676R1 

PLOS ONE

Dear Dr. Wu, 

I'm pleased to inform you that your manuscript has been deemed suitable for publication in PLOS ONE. Congratulations! Your manuscript is now being handed over to our production team.

Kind regards, 

on behalf of

Dr. Alexander Rabadi 

Academic Editor

PLOS ONE